# Inequalities in the Health Impact of the First Wave of the COVID-19 Pandemic in Piedmont Region, Italy

**DOI:** 10.3390/ijerph192214791

**Published:** 2022-11-10

**Authors:** Michele Marra, Elena Strippoli, Nicolás Zengarini, Giuseppe Costa

**Affiliations:** 1Epidemiology Department, Local Health Unit TO3, 10095 Grugliasco, Italy; 2Department of Clinical and Biological Sciences, University of Turin, 10126 Torino, Italy

**Keywords:** SARS-CoV-2, inequalities, equity access to healthcare

## Abstract

(1) Introduction: Several studies observe a social gradient in the incidence and health consequences of SARS-CoV-2 infection, but they rely mainly on spatial associations because individual-level data are lacking. (2) Objectives: To assess the impact of social inequalities in the health outcomes of COVID-19 during the first epidemic wave in Piedmont Region, Italy, evaluating the role of the unequal social distribution of comorbidities and the capacity of the healthcare system to promote equity. (3) Methods: Subjects aged over 35, resident in Piedmont on 22 February 2020, were followed up until 30 May 2020 for access to swabs, infection, hospitalization, admission to intensive care unit, in-hospital death, COVID-19, and all-cause death. Inequalities were assessed through an Index of Socioeconomic Disadvantage composed of information on education, overcrowding, housing conditions, and neighborhood deprivation. Relative incidence measures and Relative Index of Inequality were estimated through Poisson regression models, stratifying by gender and age groups (35–64 years; ≥65 years), adjusting for comorbidity. (4) Results: Social inequalities were found in the various outcomes, in the female population, and among elderly males. Inequalities in ICU were lower, but analyses only on in-patients discount the hypothesis of preferential access by the most advantaged. Comorbidities contribute to no more than 30% of inequalities. (5) Conclusions: Despite the presence of significant inequities, the pandemic does not appear to have further exacerbated health inequalities, partly due to the fairness of the healthcare system. It is necessary to reduce inequalities in the occurrence of comorbidities that confer susceptibility to COVID-19 and promote prevention policies that limit inequalities in the mechanisms of contagion and improve out-of-hospital timely treatment.

## 1. Introduction

The quick propagation of the COVID-19 epidemic, the general unpreparedness for such an unpredictable event, and the relative severity of the infection enforced the starting perception that the virus was “hitting the rich and the poor alike”, at least after stratifying by sex and adjusting by age and presence of comorbidities. Nevertheless, the implementation and prolongation of the social and physical distancing measures, as well as the increase in the consequent social and economic costs, raised new concerns about the real and wider impact of the pandemic. Many authors tried to draw explanatory frameworks aimed at identifying all the potential pathways that may have been triggered by the pandemic, and that may result in an exacerbation of pre-existing social inequalities in health [1,2,3].

Among these pathways, the causes and size of inequalities in COVID-19 morbidity and mortality have been poorly investigated [4]. Yet, clues were many. First of all, epidemiological literature of the great past and recent epidemics shows spatial and individual-based social inequalities in the incidence of infection and in mortality rates, as observed during the 1918 Spanish influenza, the 2009 H1N1 pandemic, and in many cyclical winter influenzas in Europe and in North America [5,6]. In the second line, structured and integrated frameworks drawn to depict all the mechanisms that link social disparities to health inequalities during infectious emergencies appeared to be useful also in the COVID-19 scenario. This is the case of the model developed in 2008 by Blumenshine et al. [7], which, moving from Diderichsen’s framework on the impact of the social determinants on health inequalities [8], theorized three potential sources of disparities also during influenza outbreaks: differential exposure to the virus, differential susceptibility to disease and differential access to healthcare once the disease has developed. Similarly, since the beginning of the COVID-19 pandemic, disadvantaged people in terms of income, wealth, education, occupation, gender, race, or ethnicity appeared:(a)To be more exposed to the infection because of (i) the lower level of health literacy and consequent lower compliance to rules and personal hygiene, which prevent contagion and hinder transmission, (ii) because of occupational constraints such as lower opportunities for home working and higher employment in “essential” jobs; and (iii) other reasons as higher and not avoidable use of public transports higher prevalence of living in overcrowded housings and in deprived neighborhoods where social interaction is more difficult to prevent [9,10];(b)To be more vulnerable to the disease due to the higher prevalence for almost all chronic diseases or conditions, such as obesity and unhealthy lifestyles, which are risk factors for a severe form of COVID-19 [11,12];(c)To have lower access to the health systems and to high-quality healthcare because of a generally lower level of health literacy, a potentially lower prompt detection of SARS-CoV-2 infection, and lower capability of getting out of the extraordinary legislation implemented to limit the pandemic within healthcare settings [13,14].

Indeed, early evidence suggested the existence of spatial inequalities in the distribution of COVID-19 disease. In Catalonia, Spain, municipalities with more university educated had fewer confirmed cases, and COVID-19 incidence was higher in the most deprived urban areas [15], similar to what has been found in Lombardy, one of the most affected Italian Regions [16]. In England, people who live in deprived areas have had higher diagnosis rates and more than double death rates than those living in less deprived ones [17]. Evidence of social inequalities in mortality at the census tract level has also been observed in the northern Italian Region Emilia-Romagna [18].

These spatial analyses clearly point out the existence of an equity issue in the pathway of prevention and care of COVID-19, but they are not able to disentangle the mechanisms through which social disadvantage becomes a risk factor nor to identify specific vulnerable groups to be targeted for intervention. For example, spatial analyses cannot distinguish particular working categories, such as health workers, which have had massive exposure to the virus. The first national serum prevalence survey conducted in Italy observed a two-fold prevalence of infection among doctors and nurses (5.3% vs. 2.5% in the rest of the population) [19]. Or they cannot infer if higher mortality is attributable to less time access to healthcare in more deprived areas or to a higher prevalence of underlying or susceptibility risk factors among people living there. To make this kind of interpretation, we need information at the individual level about the social characteristics and the different outcomes of COVID-19 cases. In a review of the influence of socioeconomic factors on COVID-19 transmission, severity, and outcomes, out of 29 eligible studies, only one study reported the occupational position of patients [20]. A study led in Aragon, Spain, confirmed that the probability of COVID-19 infection during the first pandemic wave was higher in more deprived areas where workers with low salaries, unemployed, and people on minimum integration income tend to live, but recognized that is necessary more research in order to attribute this effect to the area or individual inequalities [21]. More in general, health and social integrated informative systems are needed in order to evaluate the impact of individual disadvantage on health.

Recently, the Italian Institute of Statistics has shown that during the first wave in the Northern regions more affected by the pandemic educational inequalities in total mortality have slightly widened compared to the ones observed in the same months of the previous years [22].

Finally, a few studies from Sweden [23] and England [24] have explored and also confirmed the existence of an individual social gradient in the pandemic, but they focused only on a single outcome, usually mortality data, without considering multiple effects of the infection. On the contrary, the main objective of the study is to assess the impact of social inequalities on the main health outcomes of SARS-CoV-2 infection (depicting the whole health pathway from incidence and severe outcomes) using data coming from an integrated information system linking socioeconomic indicators and health outcomes at individual level in Piedmont (Northern Italy, 4.5 million inhabitants).

## 2. Materials and Methods

### 2.1. Design, Study Population, and Sources of Information

Data were drawn from the Longitudinal Study of Piedmont Region (LSP), a health monitoring system that, via individual record linkage, allows to connect of population censuses data, mortality registry, and administrative health data for all people in charge of the Regional Health Service RHS (around 4.5 million people) under the rules of the National Statistics Plan. Using anonymized keys, LSP has been integrated with data from the regional COVID-19 Platform, which traces the contacts with the RHS of individuals tested for SARS-CoV-2 and their COVID-19 outcomes [25].

Our starting study population included all the people resident in the region on 22 February 2020, the date of the first confirmed case of SARS-CoV-2 in Piedmont, who took part in the 2011 census and aged 35 or more. After that, we restricted it to the subjects who filled the “long form” questionnaire of the census (71% of the population) that, collecting data on economic activity, permitted to stratify and exclude the healthcare workers from the study (about 5% of the population) in order to avoid a selection bias. In fact, the health workers are, on average, more educated and qualified compared to the general population (Figure 1a), and, at the same time, have been regularly and more screened for infection during the first wave when the tests were poorly available and offered only to symptomatic patients in the general population (Figure 1b) [26,27]. Finally, patients who, in the 2011 census, lived in institutions were excluded, though this selection did not allow for accurately excluding people living in long-term facilities during the study period.

At the end of this process, our final study population counted 1,851,911 individuals, 51.3% of whom were females and divided into two age groups, between 35 and 64 years old (60.3%) and 65 years or older (39.7%) (Table 1).

In Figure 1b, education level has been classified as High = University Degree; Medium high = High school diploma; Medium low = Middle school or vocational school diploma; Low = Primary school or less.

### 2.2. Outcomes and Follow-Up

The follow-up lasted from 22 February 2020 to 15 May 2020, the period of the first pandemic spread in northern Italy.

The following outcomes have been investigated through the data sources:i.Access to diagnostic tests (nasopharyngeal swabs) and confirmed SARS-CoV-2 infection via the COVID-19 Platform dataset, which includes the date of the first swab and first positive test;ii.Hospitalization with a COVID-19-related diagnosis, and admission to intensive care unit (ICU)—including ventilation procedures received outside of ICUs—among infected cases using the hospitalization registries;iii.Overall deaths occurred to infected people within 30 days from the date of the first positive test, in-hospital deaths in COVID-19 patients, and all-cause mortality in the whole study population, using mortality and hospitalization registries.

Person-years (py) at risk were computed specifically for each outcome, from the baseline date to the first event between the end of follow-up.

### 2.3. Exposure and Adjustment Variables

To evaluate the impact of social inequalities, we have created a composite Index of cumulative Socioeconomic Disadvantage (ISD) which combines available information in the 2011 census, selecting variables that on one side are well-known determinants of health [28,29] and, on the other have been listed as potential predictors of the higher burden of disease within the models which have tried to identify potential sources of inequalities during respiratory pandemics [1,6,11]. In particular, we considered three dimensions: cultural, represented by the level of education; material, represented by both the quality of the dwelling and housing overcrowding; and contextual dimension (place of residence) rep, resented by an area-based deprivation index. Finally, all these dimensions were dichotomized as follows:Educational qualification in two levels, with respect to the onset of the reform of the compulsory school (introduced in the 1952 birth cohort): low for educational titles lower than primary school diploma and middle school diploma, respectively, among people born before and after 1952, high for all the higher educational titles;Overcrowding in two levels, above and below the value of the ratio between the number of inhabitants and the floor space of the dwelling that, according to the definition of Banca D’Italia is considered not sufficient [30];Housing conditions in two levels, poor and not poor, being poor housing conditions characterized by rent of small size dwellings (≤84 sqm) and dwelling with problems with inadequate heating and/or bathroom.Area deprivation in two levels, more and less deprived, being the more deprived those living in a census tract belonging to the more deprived quintile (the 5th) of an area-based deprivation index calculated that combines five dimensions of social and material deprivation (prevalence of residents in condition of low education, unemployment, home in rent, home crowding, single-parent family [31].

The final score of the ISD results as the linear sum of low educational attainment, overcrowding, poor housing conditions, and living in a highly deprived census tract. This indicator has four levels, with having 0 disadvantages being the reference category vs. having 1, 2, and 3 or 4 conditions.

The presence of comorbidities has been identified through the occurrence in the period of 2015–2019 of at least one hospitalization, drug prescriptions, or entitlement of a fee exemption for a list of medical conditions that may cause susceptibility to severe outcomes of COVID-19; the codes are reported in Table A1. Each medical condition was represented by a dichotomous variable, including diabetes, ischemic heart disease, peripheral vascular disease, congestive heart failure, cardiomyopathy, cardio-nephropathy, cerebrovascular disease, dementia and Alzheimer’s disease, peptic ulcer disease, rheumatic disease, moderate and severe liver disease, chronic pulmonary disease, hemiplegia, any malignancy, renal disease, and Human Immunodeficiency Virus.

### 2.4. Statistical Analyses

Incidence rate ratios (IRR) for each outcome (access to a diagnostic test, confirmed SARS-CoV-2 infection, hospitalization with a COVID-19-related diagnosis, deaths happened to infected people within 30 days from the date of first positive and all-cause mortality) by ISD were estimated through multivariate Poisson regression models with robust standard errors, first adjusting only for age (model 1) and then for age and comorbidities (model 2) [32]. Relative risk (RRs) of ICU and in-hospital deaths by ISD were estimated by means of Poisson regression models applying models 1 and 2 as well. All the analyses are stratified by gender and by age, comparing the results of adults (35–64 years) and the elderly (≥65 years).

To facilitate the comparison between the different outcomes, we also computed a summary measure of relative risk, the relative index of inequality (RII) for each combination of outcome and ISD, from both models 1 and 2. The RII corresponds to the ratio of the regression-based rates estimated at the two extreme points of the social hierarchy on a continuous scale, in our case, the ISD scale, representing in a single value the whole gradient (i.e., the slope) associated with each outcome.

Analyses were carried out using the STATA/SE 13 software (College Station, TX, USA).

## 3. Results

Table 1 reports the number and distribution of subjects in the study population during the first pandemic wave, as well as those affected by the seven considered outcomes by sex, age group, and ISD. During the follow-up, 71,572 tests were performed among 1,851,911 individuals (3.8% of the population), observing 12,775 positive cases (positivity rate 17.8%). In total, 40.4% of diagnosed infections have required hospitalization and 8.2% admission to the intensive care unit. Furthermore, 2509 deaths occurred among positive cases (lethality rate equals 19.6%) but only 1795 (71.5%) after hospitalization.

The proportion of outcomes by age and sex (Table A2) largely corresponds to what is known about the demographic impact of the pandemic on the whole population: the elderly (population over 65) paid a heavier burden of the disease with higher access to tests, positivity rate (19.9% of tests vs. 14.8% among the population aged from 35 to 64) and above all a much stronger lethality (28.2% vs. 3.5% of cases). Older people also showed more relevant access to hospitals (46.1% vs. 29.7% of cases), but the comparison has to consider also gender disparities: only 3 out of 10 females required hospital-centered treatments while this has been necessary among 1 out of 2 infected males; furthermore, admissions have been much more frequent among younger males than older females (40.2% vs. 34.5%). Additionally, admissions to ICU have followed a gender pattern, being more frequent among males without age differences. Finally, even deaths have been more frequent among males (24.0% vs. 15.9%).

Table 2 reports on age and comorbidity-adjusted social inequalities for the same set of outcomes. Consistent inequalities existed in the access to diagnostic tests, for which RIIs show significant social gradients in the four sex and age subgroups. This has been particularly true among older people and younger females: individuals with two or more social disadvantages in these groups presented a 50% significant excess of swab use, whereas the increase in the risk was lower among younger males, around 10%. Adjusting for comorbidity slightly decreases the size of inequalities, suggesting that the presence of chronic disease was one of the criteria considered for access to the test. Risks of infection largely reflect what has been observed about access to tests, with increasing incidence ratios according to the number of social disadvantages both in the elderly and female younger population and an uncertain social gradient among the male adult population. During the first wave, the main criterion that marked access to diagnostic tools in the general population was the presence of symptoms, while in the population of essential occupations was the periodic screening. Milder inequalities in access to infections among working-age males may be attributed to a larger proportion of males in the essential economic activities whose employees have continued to work also thanks to periodic screening.

The selection bias due to different accessibility to tests should play a lower role once we turn into COVID-19-related mortality. In particular, we expected inequalities to be significantly higher in more severe outcomes, both because of the higher prevalence in lower social groups of chronic diseases associated with unfavorable outcomes of SARS-CoV-2 infection, because of lower protection in front of the contagion and less timely access to high-quality care. Indeed, in all four subgroups, inequalities in COVID-19 mortality tend to increase, climbing up the social ladder in both genders and among the younger and older populations, with particular robustness among the elderly. Similar results, even more robust, have been observed in overall mortality (Table 2).

As expected, comorbidities play an important role, but they are not as crucial as we thought. After adjusting for the comorbidity index, RIIs in access to tests, infection rate, and mortality attenuate, especially among the younger male population and by 15–25% among the older population, where social gradients remain significant (Table 2). Therefore, the unequal prevalence of comorbidities explains only a part of the impact of social disadvantage on COVID-19 infection and mortality outcomes. There is something else out of the syndemic of chronic diseases. Gradients become not significant only among the younger population, and regarding COVID-19 mortality: the population under 65 tends to be more resilient, and deaths may occur, especially among the unluckiest few people suffering from chronic diseases.

Inequalities in hospitalization tell a similar story to mortality because hospitalization is another measure of severity in the course of the disease (Table 3). Social inequalities in hospitalization in the younger population are higher compared to the inequalities observed in infection (RII 1.37, 1.07–1.75 in men; 2.12, 1.42–3.15 in women), whereas they remain similar (and relevant) among older men (1.60, 1.36–1.89) and even decrease among women over 65 (RII from 1.84 in infection to 1.40 in hospitalization, both significant). The fact that some groups have been more tested but less hospitalized, and vice versa, may indirectly indicate that there are differences in the outpatient pathway of care that need to be investigated more in-depth.

Nevertheless, once hospitalized, admission to ICU was equally distributed—out than for younger women—and overall gradients were basically flat and reversed (even if not significantly) among older patients. This concerned us: the higher prevalence of chronic health conditions and less timely access to healthcare—made us expect higher access to ICUs among the worst off, also when hospitalized. During the first epidemic wave, few physicians started warning about the risk of being forced to implement prioritization procedures in the admission of patients because of both shortage of ICU beds and the very high healthcare demand. Had individuals with better socioeconomic status received a privileged entry to specialized care because of a better-expected prognosis? Has this caused lower access to advanced treatments within more disadvantaged social groups? Fortunately, no differences have been found in the risk of death once hospitalized, independently from the admission to ICU, disproving the hypothesis of discretionary and inequitable access to treatment. On the contrary, expected gradients have been observed among younger women, which presented relevant inequalities not only in hospitalization but also in ICU admission and in-hospital mortality. As a matter of fact, the low absolute number of outcomes in this age and gender group (only 88 adult women needed intensive treatment) leads us to be cautious in this regard.

## 4. Discussion

Relevant social inequalities in infection, hospitalization and death from COVID-19 were observed in the population of the Piedmont Region during the first wave of the COVID-19 pandemic among both adult and elderly females and among older males. In working-aged males, inequalities in infection and consequently in hospitalization and mortality were smaller, probably due to the failure to completely exclude skilled healthcare workers from the study population and because of the level of exposure to infection among skilled occupations that continued to work during the lockdown and that were regularly screened. Additionally, if the absolute number of cases was quite low among younger generations during the follow-up and obliged us to be cautious in the evaluation of the results, facts are more solid for older people: if you are more than 65 and especially if you are a female, having lower educational attainment, living in overcrowded flats or in poor housing and being resident in a deprived census tract is related to a higher probability of infection, hospitalization, and death for COVID-19.

The unequal distribution of comorbidities plays a consistent and pervasive role in explaining these inequalities, especially among younger people and for most severe outcomes. A first important policy implication, therefore, is that to avoid inequalities in this kind of pandemic (as well as to promote equity in health in general), we need to prevent the unequal epidemic of chronic diseases that cause unequal susceptibility. This means reinforcing and ensuring equity-oriented prevention plans in order to tackle the unequal exposure to behavioral and environmental risk factors as well as to include equity criteria in the care of chronic patients and to create treatment pathways capable of embracing and taking care of the different health needs from different population and social groups. Both these objectives require investments in economic as well as human resources, starting with equity capacity building for healthcare workers.

Nevertheless, adjusting for comorbidities only partially reduces the impact of social disadvantage: IRRs and RIIs for COVID-19 outcomes, even after considering the presence of chronic diseases, always remain significant (except for mortality among younger adults), and, particularly among the elderly, the gradients smooth out by only 15–25%. It is possible that our comorbidity index misclassifies the presence and, above all, the severity of chronic conditions, limiting the capacity of the variable to completely adjust the estimate of inequalities in COVID-19 outcome risk. Sensibility analyses were carried out to test the Charlson index, as defined by Quan [33] as an alternative adjusting variable (Table A3 and Table A4), concluding that the used indicator is more exhaustive than this version of Quan’s algorithm.

Nonetheless, apart from comorbidities, there is certainly more that recalls the other mechanisms contributing to inequalities in COVID-19 outcomes. Our study, however, fails to disentangle the different contributions of the individual risk factors that make up the composite index of disadvantage. Modeling attempts have been made, but results seem quite unclear and would bring us entering the realm of speculation. The fact remains, however, that in addition to the syndrome of comorbidities, there seems to be more. It is, therefore, necessary to continue investigating to better understand which the main generating pathways of inequalities are during epidemics, also in order to obtain clues and recommendations about how to include equity criteria and priorities into the current redesign of Italian national and regional plans for preparedness and response to pandemics.

Other considerations come, however, from the evaluation of the role of hospital care in COVID-19 cases. As we have seen, the system has been able to allocate equity treatment on the basis of need and to protect the equity of its patients once hospitalized. The slightly non-significant deficit of access to intensive care among the most disadvantaged elderly would not seem to depend on a presumed socioeconomic selection of patients but rather on a denominator bias, whereby the frailest elderly may have been hospitalized even when not severe, either because of the inability or lack of resources to provide effective care at home (especially when social distancing measures prevent the functionality of family support networks, which in Italy often compensate for the limits of out-of-hospital care and social services) or because of a policy of hospitalizing non-severe cases from nursing homes where there was little opportunity for isolation and capacity to manage the ill. Moreover, data on hospitalizations, intensive care, and in-hospital mortality suffer from another important bias. That is, they reflect only what arrived in the wards but say nothing about the deaths at home, in ambulances, and in nursing homes of patients who would have needed treatment but did not arrive in time to the healthcare facilities. The epidemiologic literature is replete with examples showing less access for less fortunate segments of the population, and it is likely that here too, there was a social gradient in non-arrival that, in part, may have fostered the absence of revealed inequalities in hospital indicators. Indeed, non-hospitalized deaths, 30% of the total, accounted for only 20% of deaths among the more advantaged but over 35% of the most disadvantaged deaths. In any case, considering the other gradients in 30-day mortality among the elderly not explained by comorbidities, it is likely that investment should be made to promote more equitable and timely access to the healthcare system and to improve the quality of territorial healthcare.

All these hypotheses suggest other further implications for tackling inequalities and decreasing demand on hospitals during future epidemics but, again, also in normal times: investing in primary and territorial healthcare as well as in home care, with offers attentive to the different needs of different segments of the population and, of course, taking on the age-old problem of the quality of treatment in nursing homes and in residential An implication for research is that this outpatient pathway should be further explored as for mechanisms generating potential inequalities.

Finally, we should consider three important limitations of our study that may help to better interpret the results and their implications for policies.

The size of households and the quality of the housing may overestimate the impact of social inequalities. As we know, living in an overcrowded flat or in numerous families may increase the risk of exposure and of infection. If we were evaluating the impact of non-communicable diseases, this would not be a problem, as each event may be attributed to exposure to the risk factor. Estimating the impact of a transmissible infection is different, as multiple contagions within the same family may be caused not by an exogenous higher exposure to the virus but by an endogenous and relatively higher proximity to (more) relatives. Additionally, given the fact that overcrowding and poor housing conditions are material conditions associated with low socioeconomic status (as a matter of fact, they have been used in our index of social disadvantage), this may overestimate the real impact of inequalities. In our study, identifying the first contagion in each family and using his/her social position to estimate inequalities in the incidence of the disease or in other health outcomes may have been a solution, but it has not been possible, but even if feasible, totally removing this effect may be arguable. Imagine, for example, an adolescent not respecting home confinement because of low knowledge or scarce compliance to the rules (that may be associated with low education) or consider a middle-aged man working as an essential worker on a farm (and having a low social status) or not having smart working as an option: their higher risk of exposure to the infection is surely facilitated by the consequences of their social position but so is the potential source of contagion of households member, because of overcrowding of deprived families and their highly similar social composition. Working with first households’ cases will therefore allow discriminating between the direct and indirect impact of social disadvantage on health, but one should not forget that both of these mechanisms should be considered when evaluating the overall burden of disease attributable to social inequalities.

A second and close issue is related to the failure of the study to identify residents in long-term care facilities in our sample population. After the first month of the pandemic, in April 2020, the virus invaded nursing homes (more than one thousand in the Piedmont Region), which resulted in being widely unprepared to tackle the spread of the virus among their residents. In total, 27.5% of overall cases in the general population have been diagnosed among people living in these venues during the first epidemic wave in Piedmont [34]. Many facilities started offering first treatments to their infected guests (and to other old people transferred from smaller facilities) but were unable to avoid contagion of other patients and of caregivers and health workers. Many elderlies died directly in their bed without having time to be transferred to hospitals. Beyond the tragedy, this situation may have exogenously affected inequalities, as the population of nursing homes tends to have a slightly more disadvantaged social profile than people of the same age living in households. In this sense also, the higher IRRs observed in, the older population may be an overestimated consequence of the combination of higher individual risk and propensity to live in socially segregated long-term facilities. However, is this segregation not one of the dimensions and consequences of low socioeconomic position?

Finally, the last issue stems from the methodological consequences of using information from the 2011 census to measure individual exposure to social covariates. This was a forced decision since this source is the most updated as well as the unique one with available data at the regional level. It does not affect our estimate of educational attainment, as our study population is composed of individuals with ages equal to or over 35 at the beginning of our follow-up. They had at minimum 26 years old in 2011, and they had almost surely already completed their education by that date (out of very marginal cases, especially considering our dichotomous threshold). On the contrary, deprivation of the neighborhood of residence, overcrowding, and housing conditions may have varied over a decade because of urban transformations, changes in residence, or in household composition. Estimating the precise impact of this potential distortion is a not easy task that may be partially revealed only when data coming from the next census wave, which is ongoing, will be available. Nevertheless, we are rather confident that the bias, if any, may be minor and not change the direction and, above all, the meaningfulness of the results. First, preliminary analyses have been carried out evaluating the impact of educational level as a unique social covariate on SARS-CoV-2 infection, COVID-19 hospitalization and mortality and overall mortality during the first epidemic wave in Piedmont Region (Table A5, in Appendix D). Although there are some differences in the intensity of the effects, overall results are consistent with data presented in Table 2 but seem statistically less stable and more difficult to understand in specific educational categories with very low numerators and denominators (as in younger females). Using a more complex socioeconomic indicator has sharpened our results, showing a social gradient according to the number of social disadvantages and including inequality dimensions which have been listed as predictors of unequal exposure to infectious diseases in pandemics [7,35]. It is quite unlikely that this social gradient may be the result not of having built a more sensitive index but of a bias in the estimation of the other time-dependent variables, also because we do not have clues to argue a socially different pattern in trends of the exposure to our social covariates. Partial misclassification may have occurred but in a non-differential way so that the overall effect may have diluted and underestimated the effect of social inequalities. Finally, this misclassification may affect a marginal share of the population, or at least this seems to be the case in the temporal changes in the quintile distribution of the deprivation of census tract of residence. Sensitivity analyses for this indicator have been possible for people living in Turin, the capital of the Piedmont Region, for which changes of residence can be reconstructed to the present day. From 2011 to 2020, only 18.0% of the population living in Turin on 1 January 2011 changed quintile in the distribution of its own census tract, and only 6.9% changed category in the dichotomic variable used in our study to measure deprivation level (Appendix E, Table A6 and Table A7).

## 5. Conclusions

As much as the SARS-CoV-2 virus might be expected to be blind to socioeconomic position, infection and disease course were highly unequal in the first wave of the COVID-19 pandemic, at least in the Piedmont Region of Italy, affecting the most disadvantaged social groups more often and more harshly.

The fact that disparities in severe outcomes are greater than those in infection suggests that the different susceptibility caused by the unequal social distribution of chronic diseases is one of the most relevant mechanisms in delineating these inequalities, as moreover confirmed by the attenuation of observed inequalities in hospitalization and mortality when adjusted for comorbidity measures. Tackling inequalities in these health conditions is therefore a first public health priority. However, this attenuation is only moderate, and although our study was unable to distinguish the individual contributions of the four dimensions considered in our social disadvantage index, the estimates associated with overall disadvantage remain statistically significant even after adjustment for comorbidity. Potential bias coming from using data from the 2011 census to evaluate time-dependent variables may partially underestimate inequalities, and it suggests we be cautious in our claims, but it seems that there is much more than the syndemic of chronic diseases.

Pending further evidence on this issue, a special equity lens should be included in the design of upcoming pandemic preparedness plans in order to avoid the risk of exacerbating inequalities during pandemics and to at least reduce the avoidable component of the burden of disease borne by the healthcare system.

This is also because although the healthcare system has proven to be equitable in providing care to all people once they are admitted to the hospital, regardless of social position, much demand has not been met, especially at the expense of the most disadvantaged groups of the population. In addition to increasing access, it is also crucial to invest in the equity of out-of-hospital and community-based healthcare.

## Figures and Tables

**Figure 1 ijerph-19-14791-f001:**
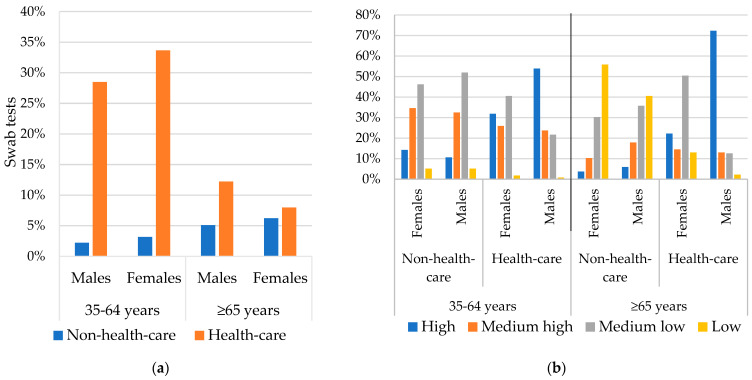
Comparison between healthcare workers and non-healthcare workers regarding (**a**) access to swabs by age and gender; (**b**) distribution of population according to educational level, by age and gender individual Index of Socioeconomic Disadvantage (ISD).

**Table 1 ijerph-19-14791-t001:** Distribution of outcomes by individual Index of Socioeconomic Disadvantages (ISC) among men and women, aged 34–64 and over 65.

Sex/ Age	Index of Social Disadvantage	Population		Swab Tests		Cases		Hospitalization		ICU Hospitalization		In-Hospital Mortality		COVID-19 Deaths		Overall Deaths	
		N	%	N	%	N	N	N	%	N	%	N	%	N	%	N	%
Males 35–64	0 disadvantages	174,858	30.4%	3676	28.9%	728	31.5%	244	26.2%	74	26.1%	25	22.9%	25	20.5%	126	20.5%
	1 disadvantage	251,236	43.6%	5448	42.9%	1001	43.3%	428	46.0%	127	44.9%	50	45.9%	59	48.4%	297	48.2%
	2 disadvantages	97,043	16.9%	2270	17.9%	360	15.6%	155	16.7%	48	17.0%	24	22.0%	27	22.1%	122	19.8%
	3 or 4 disadvantages	52,550	9.1%	1317	10.4%	224	9.7%	103	11.1%	34	12.0%	10	9.2%	11	9.0%	71	11.5%
	Total	575,687		12,711		2313		930		283		109		122		616	
Females 35–64	0 disadvantages	184,554	34.1%	4848	28.5%	587	27.9%	94	24.7%	13	14.9%	5	18.5%	10	31.3%	71	22.7%
	1 disadvantage	222,682	41.2%	7026	41.3%	878	41.8%	168	44.1%	41	47.1%	7	25.9%	10	31.3%	143	45.7%
	2 disadvantages	86,265	16.0%	3231	19.0%	388	18.5%	66	17.3%	19	21.8%	8	29.6%	6	18.8%	62	19.8%
	3 or 4 disadvantages	47,232	8.7%	1909	11.2%	248	11.8%	53	13.9%	14	16.1%	7	25.9%	6	18.8%	37	11.8%
	Total	540,733		17,014		2101		381		87		27		32		313	
Males 65+	0 disadvantages	132,647	40.6%	5353	32.1%	1213	33.8%	761	34.4%	204	42.1%	330	32.4%	401	31.0%	1662	29.4%
	1 disadvantage	143,740	43.9%	7855	47.2%	1654	46.0%	1003	45.4%	187	38.6%	466	45.7%	613	47.3%	2814	49.8%
	2 disadvantages	38,922	11.9%	2617	15.7%	523	14.6%	313	14.2%	68	14.0%	162	15.9%	212	16.4%	884	15.7%
	3 or 4 disadvantages	11,792	3.6%	826	5.0%	202	5.6%	132	6.0%	25	5.2%	61	6.0%	69	5.3%	286	5.1%
	Total	327,101		16,651		3592		2209		484		1019		1295		5646	
Females 65+	0 disadvantages	121,316	29.7%	4857	19.1%	883	18.5%	377	22.9%	61	30.0%	121	18.9%	181	17.1%	1058	17.3%
	1 disadvantage	209,429	51.3%	13986	55.1%	2610	54.7%	882	53.7%	95	46.8%	356	55.6%	578	54.5%	3466	56.7%
	2 disadvantages	60,449	14.8%	5058	19.9%	979	20.5%	280	17.0%	39	19.2%	124	19.4%	230	21.7%	1247	20.4%
	3 or 4 disadvantages	17,196	4.2%	1475	5.8%	297	6.2%	104	6.3%	8	3.9%	39	6.1%	71	6.7%	338	5.5%
	Total	408,390		25,376		4769		1643		203		640		1060		6109	

Data are presented stratified by gender and age.

**Table 2 ijerph-19-14791-t002:** Access to swabs, SARS-CoV-2 confirmed infection, and for COVID-19 and all-causes mortality incidence rate ratios (RR) and Relative Index of Inequality (RII) by index of socioeconomic disadvantage (ISD), gender and age group.

	ISD	Swab Tests			Infection			COVID-19 Mortality			Overall Mortality		
		IRR	95% CI		IRR	95% CI		IRR	95% CI		IRR	95% CI	
**Males 35–64 years**	**0 disadvantages**	1.00			1.00			1.00			1.00		
	**1 disadvantage**	1.01	0.97	1.05	0.91	0.83	1.00	1.44	0.90	2.31	**1.45**	**1.18**	**1.79**
	**2 disadvantages**	**1.10**	**1.05**	**1.16**	**0.88**	**0.77**	**0.99**	**1.87**	**1.09**	**3.23**	**1.68**	**1.31**	**2.16**
	**3–4 disadvantages**	**1.20**	**1.13**	**1.28**	**1.04**	**0.89**	**1.21**	1.56	0.77	3.16	**1.98**	**1.48**	**2.65**
	**RII ^1^**	**1.20**	**1.13**	**1.28**	0.92	0.78	1.07	**2.10**	**1.09**	**4.05**	**2.21**	**1.65**	**2.98**
	**RII adjusted ^2^**	**1.11**	**1.04**	**1.19**	0.87	0.74	1.01	1.57	0.79	3.12	**1.73**	**1.27**	**2.35**
**Females 35–64 years**	**0 disadvantages**	1.00			1.00			1.00			1.00		
	**1 disadvantage**	**1.26**	**1.21**	**1.30**	**1.22**	**1.10**	**1.36**	0.65	0.27	1.58	**1.32**	**1.00**	**1.76**
	**2 disadvantages**	**1.47**	**1.40**	**1.53**	**1.40**	**1.24**	**1.60**	1.12	0.41	3.06	**1.64**	**1.16**	**2.30**
	**3–4 disadvantages**	**1.55**	**1.47**	**1.63**	**1.65**	**1.42**	**1.92**	2.32	0.84	6.39	**2.02**	**1.36**	**3.00**
	**RII ^1^**	**1.76**	**1.67**	**1.86**	**1.78**	**1.52**	**2.09**	2.05	0.43	9.85	**2.27**	**1.47**	**3.50**
	**RII adjusted ^2^**	**1.69**	**1.60**	**1.78**	**1.69**	**1.44**	**1.98**	1.46	0.30	7.06	**1.86**	**1.20**	**2.89**
**Males ≥ 65 years**	**0 disadvantages**	1.00			1.00			1.00			1.00		
	**1 disadvantage**	**1.20**	**1.16**	**1.25**	**1.12**	**1.04**	**1.21**	**1.13**	**1.00**	**1.29**	**1.21**	**1.14**	**1.29**
	**2 disadvantages**	**1.54**	**1.47**	**1.61**	**1.35**	**1.22**	**1.50**	**1.51**	**1.28**	**1.79**	**1.47**	**1.36**	**1.60**
	**3–4 disadvantages**	**1.74**	**1.62**	**1.87**	**1.87**	**1.61**	**2.17**	**1.84**	**1.43**	**2.37**	**1.81**	**1.59**	**2.05**
	**RII ^1^**	**1.82**	**1.72**	**1.93**	**1.62**	**1.43**	**1.85**	**1.78**	**1.43**	**2.22**	**1.79**	**1.61**	**1.99**
	**RII adjusted ^2^**	**1.65**	**1.56**	**1.75**	**1.46**	**1.28**	**1.65**	**1.58**	**1.27**	**1.96**	**1.59**	**1.43**	**1.76**
**Females ≥ 65 years**	**0 disadvantages**	1.00			1.00			1.00			1.00		
	**1 disadvantage**	**1.21**	**1.17**	**1.25**	**1.23**	**1.14**	**1.33**	**1.23**	**1.04**	**1.46**	**1.21**	**1.13**	**1.30**
	**2 disadvantages**	**1.44**	**1.39**	**1.50**	**1.52**	**1.38**	**1.66**	**1.58**	**1.30**	**1.93**	**1.39**	**1.28**	**1.52**
	**3–4 disadvantages**	**1.63**	**1.54**	**1.72**	**1.77**	**1.55**	**2.02**	**1.91**	**1.45**	**2.52**	**1.49**	**1.32**	**1.69**
	**RII ^1^**	**1.69**	**1.61**	**1.78**	**1.84**	**1.64**	**2.07**	**2.00**	**1.55**	**2.57**	**1.58**	**1.43**	**1.76**
	**RII adjusted ^2^**	**1.57**	**1.50**	**1.65**	**1.69**	**1.51**	**1.91**	**1.81**	**1.40**	**2.33**	**1.45**	**1.30**	**1.60**

^1^ Model adjusted for age (continuous) ^2^ Model adjusted for age (continuous) and comorbidities.

**Table 3 ijerph-19-14791-t003:** Incidence of hospitalization and ICU admission rate ratios (IRR), Risk of ICU admission and in-hospital mortality Ratio (RR) among hospitalized people and Relative Index of Inequality (RII) for the index of social disadvantage (ISD) by gender and age group.

	ISD	Hospitalization			ICU Admission			ICU among Hospitalized Cases			In-Hospital Mortality among Hospitalized Cases		
		IRR	95% CI		IRR	95% CI		IRR	95% CI		IRR	95% CI	
**Males 35–64 years**	**0 disadvantages**	1.00			1.00			1.00			1.00		
	**1 disadvantage**	1.11	0.95	1.30	1.07	0.81	1.43	0.96	0.75	1.22	1.07	0.68	1.68
	**2 disadvantages**	1.11	0.91	1.36	1.13	0.79	1.62	0.99	0.73	1.34	1.40	0.83	2.36
	**3–4 disadvantages**	**1.46**	**1.16**	**1.83**	**1.60**	**1.06**	**2.40**	1.10	0.79	1.53	0.98	0.49	1.95
	**RII ^1^**	**1.37**	**1.07**	**1.75**	1.48	0.94	2.32	1.07	0.74	1.54	1.29	0.67	2.48
	**RII adjusted^2^**	**1.25**	**0.97**	**1.59**	1.33	0.84	2.09	1.10	0.76	1.59	0.89	0.45	1.78
**Females 35–64 years**	**0 disadvantages**	1.00			1.00			1.00			1.00		
	**1 disadvantage**	1.23	0.95	1.59	**2.19**	**1.16**	**4.13**	1.77	0.99	3.17	0.77	0.25	2.36
	**2 disadvantages**	1.35	0.99	1.85	**2.82**	**1.39**	**5.74**	**2.09**	**1.11**	**3.95**	2.26	0.78	6.53
	**3–4 disadvantages**	**2.19**	**1.56**	**3.06**	**4.18**	**1.96**	**8.90**	1.90	0.97	3.73	2.50	0.83	7.55
	**RII ^1^**	**2.12**	**1.42**	**3.15**	**5.13**	**2.27**	**11.58**	**2.17**	**1.14**	**4.14**	**4.82**	**1.04**	**22.42**
	**RII adjusted ^2^**	**1.81**	**1.22**	**2.70**	**4.19**	**1.83**	**9.59**	**2.27**	**1.14**	**4.52**	4.44	0.98	20.11
**Males ≥ 65 years**	**0 disadvantages**	1.00			1.00			1.00			1.00		
	**1 disadvantage**	**1.11**	**1.01**	**1.22**	0.87	0.71	1.06	**0.79**	**0.67**	**0.94**	0.96	0.87	1.06
	**2 disadvantages**	**1.31**	**1.15**	**1.50**	1.16	0.88	1.52	0.93	0.74	1.17	1.05	0.92	1.20
	**3–4 disadvantages**	**1.96**	**1.63**	**2.35**	1.38	0.91	2.09	0.81	0.57	1.17	0.93	0.77	1.13
	**RII^1^**	**1.60**	**1.36**	**1.89**	1.10	0.78	1.57	0.77	0.58	1.03	1.00	0.84	1.18
	**RII adjusted ^2^**	**1.46**	**1.24**	**1.72**	1.01	0.71	1.42	0.78	0.58	1.04	0.96	0.81	1.13
**Females ≥ 65 years**	**0 disadvantages**	1.00			1.00			1.00			1.00		
	**1 disadvantage**	**1.10**	0.97	1.25	0.90	0.65	1.25	0.81	0.60	1.09	1.11	0.94	1.30
	**2 disadvantages**	**1.17**	**1.00**	**1.37**	1.29	0.85	1.94	1.06	0.73	1.53	**1.22**	**1.00**	**1.48**
	**3–4 disadvantages**	**1.63**	**1.31**	**2.03**	0.93	0.44	1.94	0.55	0.27	1.12	1.07	0.81	1.42
	**RII ^1^**	**1.40**	**1.15**	**1.70**	1.20	0.69	2.08	0.83	0.51	1.35	1.22	0.97	1.53
	**RII adjusted ^2^**	**1.24**	**1.02**	**1.51**	1.02	0.58	1.78	0.81	0.49	1.32	1.19	0.94	1.50

^1^ Model adjusted for age (continuous) ^2^ Model adjusted for age (continuous) and comorbidities.

## Data Availability

Not applicable.

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
