# Peer review of "Inequalities in the Health Impact of the First Wave of the COVID-19 Pandemic in Piedmont Region, Italy"

_ijerph, 2022, doi:10.3390/ijerph192214791_

Round 1

Reviewer 1 Report

Thank you very much for sending me this manuscript. This is a very interesting article. I have several minor comments.

1. Can you briefly discuss your variable selection? Any framework? Is it worthwhile mentioning something like social determinants of health framework?

2. Would you be able to provide potential policy implications based on your interesting findings?

3. There are some typos throughout the manuscript. Please carefully check them.

Author Response

  1. Can you briefly discuss your variable selection? Any framework? Is it worthwhile mentioning something like social determinants of health framework?

We have mentioned the conceptual framework elaborated by Blumenshine et al which, moving from the Diderichsen’s framework on the impact of the social determinants on health inequalities, theorized three potential sources of disparities during influenza outbreaks and respiratory pandemics.

We have also presented these sources and evaluated how they do fit with Covid-19 scenario (page 2, Introduction).

We have discussed more how we built our Index of cumulative Social Disadvantage (ISD), explaining the selection of our single variables (page 5, Materials and methods)

  1. Would you be able to provide potential policy implications based on your interesting findings?

Few policy implications were already present in the study in the “Conclusions” paragraph. We have moved them into the “Discussion” paragraph and we have deepened their discussion. We have also rewritten the Conclusion paragraph.

  1. There are some typos throughout the manuscript. Please carefully check them.

Ok

Reviewer 2 Report

Comment

1. Inequalities in the health impact of the first wave of the pandemic in Piedmont Region, Italy.

2. The manuscript shows an interesting theme. Studies generally show that certain adverse health scenarios are accentuated by inequities and inequalities. As the researchers indicate, there are few studies that analyze these issues at an individual level.

3. It is considered that the proposed design is adequate, and the limitations shown show the need to propose more complex designs to address the different scenarios faced by the populations. I suggest for future work the use of multilevel regressions.

Author Response

  1. Inequalities in the health impact of the first wave of the pandemic in Piedmont Region, Italy.
  2. The manuscript shows an interesting theme. Studies generally show that certain adverse health scenarios are accentuated by inequities and inequalities. As the researchers indicate, there are few studies that analyze these issues at an individual level.
  3. It is considered that the proposed design is adequate, and the limitations shown show the need to propose more complex designs to address the different scenarios faced by the populations. I suggest for future work the use of multilevel regressions.

We kindly thank the reviewer for the suggestion and we will consider it for future analysis. We intend that the reviewer is recommending considering also a “household” level, which is a promising approach especially in order to distinguish between endogeneous/directs and exogenous/indirect effects of individual social disadvantage. Nevertheless this approach was not so applicable in our study as we should rely even more on 2011 census data which as observed by another review, may have changed over time. In particular, household composition may have varied. But it is an important idea that we will try to deepen in future analyses (and above all when studying also other Covid-19 waves, as we focused only on the first one).

Reviewer 3 Report

I have read the manuscript with interest and congratulate you on your study. Still, I would like to make some comments and suggestions.

In the introduction the authors mention several studies carried out in regions of European countries. It is suggested that the same criteria for localisation should be followed in all cases. Aragon is mentioned as belonging to Spain, but Calalonia is not. The same is true for Lombardy vs. Piedmont.

The number of the participants in the study is not known until the results section (table 1). Please describe the sample in the previous section.

It is not clear when the socio-economic data is obtained - is it in 2011? This is the most relevant issue, as it may call into question all your results.

Author Response

  1. In the introduction the authors mention several studies carried out in regions of European countries. It is suggested that the same criteria for localisation should be followed in all cases. Aragon is mentioned as belonging to Spain, but Calalonia is not. The same is true for Lombardy vs. Piedmont.

Harmonised

  1. The number of the participants in the study is not known until the results section (table 1). Please describe the sample in the previous section.

We have described our sample at the end of the study population paragraph within material and methods section. We have also added a starting paragraph (and a supplementary table, in a new appendix) in the results section presenting better descriptive results for all our sample

  1. It is not clear when the socio-economic data is obtained - is it in 2011? This is the most relevant issue, as it may call into question all your results.

We have added and discussed this limit in the discussion section

Round 2

Reviewer 3 Report

I thank the authors for their clarifications.

In table 1 and table B, the two age groups appear between 35 and 64 years or greater than or equal to 65. The text repeatedly refers to the 30-64 age group.

Figure 1 is not referenced in the text.

The limitations of the study should appear in the conclusions section, not in the discussion of the results.

I remain unconvinced by the use of the 2011 census data. Indeed, variables may change over time and may do so differentially in different age groups. The authors speak of a preliminary study for which no specific data are given and it is not known whether it has been published. Moreover, what do they mean by consistent results if one study uses only one social covariate and another uses four, or do the other variables contribute nothing to the research?  I recommend that you present the results of the preliminary study in this manuscript, in case it is not published.

Author Response

  1. In table 1 and table B, the two age groups appear between 35 and 64 years or greater than or equal to 65. The text repeatedly refers to the 30-64 age group.

Thank you very much for this clarification. The correct age range is 35-64 years. We have changed the error throughout the text

  1. Figure 1 is not referenced in the text.

Done

  1. The limitations of the study should appear in the conclusions section, not in the discussion of the results.

We have been checking in various IJERP papers and limitations are always discussed in the discussion section. Furthermore, the conclusion section is always a very brief section containing the main results of the study. We would prefer to keep it as it is now. By the way, we have added a sentence on the main limitation of our study even in the conclusion section.

  1. I remain unconvinced by the use of the 2011 census data. Indeed, variables may change over time and may do so differentially in different age groups. The authors speak of a preliminary study for which no specific data are given and it is not known whether it has been published. Moreover, what do they mean by consistent results if one study uses only one social covariate and another uses four, or do the other variables contribute nothing to the research? I recommend that you present the results of the preliminary study in this manuscript, in case it is not published.

We are aware that the use of data from the 2011 census may be a limitation to our analysis and, in fact, we have eplicted this in the discussion. However, we believe that any bias may be minor and does not call into question the significance of our results. We discuss this extensively in the last paragraph of the discussion session, where we have also included (and discussed) the results of two additional appendices.

As suggested, we have added the results of our preliminary (and unpublished) analysis of the impact of educational level on the risk of infection, hospitalization and mortality from COVID-19 and overall mortality (Appendix D).

In addition, we also included a second analysis (Appendix E) on how the distribution of the deprivation index changed from 2011 to 2020 and, in particular, how the distribution of quintiles changed, showing that the share of the population that changed classification in the dichotomous deprivation variable we used in our analysis is 6.9 percent.